**Data Availability Statement:** All relevant data are within the article and its Supporting Information files.

**Funding:** The authors received no specific funding for this work.

# COVID-19 infection among vaccinated and unvaccinated: Does it make any difference?

**Samar Fatima**[ID][1]*, **Amara Zafar**[1], **Haris Afzal**[1], **Taymmia Ejaz**[ID][1], **Sara Shamim**[ID][1], **Shayan Saleemi**[1], **Amna Subhan Butt**[2]

1 Department of Medicine, Section of Internal Medicine, Aga Khan University Hospital, Karachi, Pakistan,
2 Department of Medicine, Section of Gastroenterology, Aga Khan University Hospital, Karachi, Pakistan

* samar.fatima@aku.edu

## Abstract

### Objective

There is a probability that vaccination may lead to reduction in the severity and complications associated with COVID-19 infection among hospitalized patients. This study aimed to determine the characteristics, clinical profiles, and outcomes of COVID-19 infection in vaccinated and non-vaccinated patients.

### Design and data sources

This prospective observational cohort study was conducted at the Aga Khan University Hospital (AKUH) and recruited COVID-19 patients admitted between June 1st and September 30th, 2021. Patients' demographics, date of admission and discharge, comorbid conditions, immunization status for COVID-19 infection, presenting complaints, lab workup and computed tomography (CT) scan findings were obtained from the medical records. The primary outcome of the study was patients' condition at discharge and the secondary outcomes included level of care, length of stay (LOS), requirement of non-invasive ventilation (NIV) and inotropic support.

### Results

Among a cohort of 434 patients, 37.7% (n = 164), 6.6% (n = 29) and 55.5% (n = 241) were fully vaccinated, partially vaccinated, and unvaccinated, respectively. Around 3% and 42.9% of the patient required inotropic and NIV support respectively; however, there was no discernible difference between them in terms of vaccination status. In case of unvaccinated patients there were significantly increased number of critical care admissions (p-value 0.043). Unvaccinated patients had significantly higher median serum procalcitonin, ferritin, LDH and D-dimer levels. Around 5.3% (n = 23) of the patient required invasive ventilation and it was more common in unvaccinated patients (p-value 0.04). Overall, mortality rate was 12.2% (n = 53) and this was higher (16.2%, n = 39) in unvaccinated patients as compared to fully vaccinated patients (6.1%, n = 10, p-value 0.006).

**Competing interests:** The authors have declared that no competing interests exist.

## Conclusions

Through this preliminary data, we can conclude that patient can develop severe and critical COVID-19 infection despite being vaccinated but this proportion is low as compared to unvaccinated population. So, uninterrupted endeavors need to be done to vaccinate as many individuals as possible. Furthermore, more effective vaccinations need to be developed to lessen the high death toll of COVID-19 infection.

## Introduction

Since the start of the pandemic, COVID-19 infection has resulted in 9.4 million deaths worldwide resulting in one of the major global health crises of the 21st century [1]. Hence, the tremendous amount of efforts led to the unprecedented development and manufacturing of SARS-CoV-2 vaccines, along with their rapid approvals for emergency use and marketing authorizations, and that happened at an unparalleled pace [2]. As of March 2022, an estimated 10 billion vaccine doses have been administered worldwide and at least one dose of vaccine has been administered to 63.1% of the global population [3].

In Pakistan by the end of February 2022, 1,505,328 cases of COVID-19 with 30,114 deaths were reported [4]. With the timely efforts from the government of Pakistan an estimated 215 million vaccine doses have been administered and ninety-nine million (44.8%) of the population has been fully vaccinated. Currently, six types of COVID-19 vaccines have been approved in Pakistan which are Sinopharm (BBIBP-CorV), CanSino (AD5-nCOV), Sinovac (PiCoVacc), Sputnik (Gam-COVID-Vac), Pfizer (BNT162b2), and AstraZeneca (AZD1222, ChAdOx1 nCoV-19).

In a meta-analysis of 51 studies across 14 countries on vaccine effectiveness by Zheng et al. [5], vaccine effectiveness against infection, hospitalization, ICU admission, and mortality was 89.1%, 97.2%, 97.4%, and 99%, respectively. However, there are still certain scientific concerns to be answered, such as duration of vaccine effectiveness, vaccination regimens, and the need for booster doses. Most studies have been conducted on the real-world effectiveness of the vaccine in preventing infection and hospitalization. However, breakthrough infections can occur as no vaccine is 100% effective and outcomes of patients with breakthrough infections requiring hospitalization also need to be studied. While long-term studies need to be done on vaccine effectiveness, studies on breakthrough infections can help us understand the nature and course of this illness among vaccinated individuals and guide us in public health preparedness. Considering the geographical variability of human responses and environmental factors in various regions, the information regarding the effectiveness of vaccination in preventing progression to severe and critical illness leading to admission in intensive care unit and the impact of vaccination on characteristics, clinical profiles, and outcomes of COVID-19 infection is lacking from many regions, especially from low or middle-income countries. To the best of our knowledge, no data has been reported on outcomes of hospitalized COVID-19 patients based on vaccination status from Pakistan and very few studies are available from South Asia addressing the same issue.

Therefore, this study aimed to determine the characteristics, clinical profiles, and outcomes of COVID-19 infection in vaccinated and non-vaccinated hospitalized patients and to evaluate the impact of vaccination in preventing the severity of infection in one of the largest tertiary care centers in Pakistan.

## Material and methods

### Study design/Data source

This prospective observational cohort study was conducted at the Aga Khan University Hospital (AKUH). AKUH is one of the largest JCIA (Joint Commission International Accreditation) accredited tertiary care hospitals in Pakistan and caters to a diverse group of patients from all over the country. In February 2020, when we received Pakistan's first COVID case, our hospital executed a COVID-19 preparedness plan, anticipating increased COVID-19 patient turnover. A unit was dedicated within the hospital for suspected and confirmed COVID-19 patients. These units included (ICU), high dependency unit (HDU), and wards with an accommodation capacity of 120 beds in the COVID-19 unit.

### Eligibility criteria and data collection

The study population included patients above 18 years of age admitted between 1st June and 30th September 2021. Hospitalized patients with at least one positive SARS-CoV-2 reverse transcription polymerase chain reaction (RT-PCR) result on nasopharyngeal/oropharyngeal swab or tracheal sample were included in the study. However, patients with symptoms suggestive of COVID-19 infection but with negative SARS-CoV-2 RT-PCR results were excluded from the study. A total of 884 patients were admitted with confirmed COVID-19 infection during this period. After excluding the patients with missing data, a total of 434 confirmed cases admitted to ICU, HDU, and wards were analyzed for this study.

According to the National Institute of Health (NIH) Pakistan guidelines [6], the severity of the disease was graded as asymptomatic/non-severe, severe, and critical. Asymptomatic/non-severe disease was defined as when the patient tested positive for SARS-CoV-2 using a virologic test but did not have any symptoms or had only mild symptoms with SpO2 of greater than 94% on room air. The individuals who had SpO2 of less than 94% on room air, a ratio of arterial partial pressure of oxygen to fraction of inspired oxygen (PaO2/FiO2) of <300 mm Hg, a respiratory rate >30 breaths/min, or lung infiltrates >50% were labeled as having severe disease. However, the patient was considered as having critical disease when they had respiratory failure requiring invasive or noninvasive ventilation, septic shock, and/or multiple organ dysfunction.

According to their vaccination status, the patients were classified into three groups: fully vaccinated, partially vaccinated, and unvaccinated. The individuals were defined as "fully vaccinated" when they became symptomatic after 14 days of receiving the second dose of the vaccine. However, the "partially vaccinated" individuals were defined as patients who became symptomatic two or more weeks after the first dose or didn't receive the second dose. Patients who became symptomatic in less than 2 weeks after receiving the second dose were also labelled as partially vaccinated. "Unvaccinated individuals" were defined as those who had not received any vaccine dose [7].

Based on the percentage involvement, individual lobar scoring was done, score ranged 1–5 respectively for involvement 5% or less, 5–25%,26%-49%,50–75% and >75% involvement. CT severity scoring was based on sum of all the individual lobar scores and categorized as Mild, score ≤7; 8–17 Moderate and ≥18 as Severe [8].

Patient's demographics, date of admission and discharge, comorbid conditions, immunization status for COVID-19, presenting complaints, laboratory parameters and computed tomography (CT) scan findings were reviewed. Type of vaccination, duration of onset of symptoms after vaccination, and the number of doses of vaccine administered were also noted. The primary outcome was the status of the patient at discharge and categorized as the

"survived and discharged", "LAMA" (Leave Against Medical Advice), and "expired". Level of care, requirement of non-invasive ventilation (NIV)/inotropic support and the length of hospital stay (LOS) were all secondary outcomes.

### Patient and public involvement statement

This was a prospective observational study conducted by reviewing medical charts and electronic data. There was no live interview or direct interaction with the patients in this study. Patients' confidentiality and anonymity were maintained, no identifiers that can be used to track participants were utilized, and the research questionnaire was identified by a serial number. The study was approved as an exemption by the ethical review committee (ERC) of Aga Khan University Hospital (AKUH), Karachi, Pakistan (IRB reference number: 2021-6478-18343).

### Statistical analysis

The data were entered and analyzed using the Statistical Package for Social Science (SPSS) version 23. Results were presented as mean ± standard deviation or median with interquartile range (IQR) for continuous variables and number (percentage) for categorical variables. Analytical analysis was done according to the study objectives. For comparative analysis, Chi-square, or Fischer to exact for categorical variables and Mann–Whitney U, or independent sample t-test wherever applicable. All p-values were two-sided and considered as statistically significant if < 0.05. Univariate and multivariate regression analysis (Odds ration [OR] with 95% confidence interval [CI]) was performed to determine independent predictors of mortality in patients.

## Results

### Baseline and clinical characteristics of study subjects

A total of 434 admissions were included in the study. The median (IQR) age of the cohort was 61 years (range 44–70). The majority (54.1%) were males, and the most common comorbid condition of the study population was hypertension (49.3%), followed by diabetes mellitus (43.8%). Approximately 28.1% of the study participants had $\geq 2$ comorbid conditions. The most common presenting symptoms at the time of admission were fever (79.7%) and dyspnea (68.9%), followed by cough (60.8%). The median (IQR) CALL score of the cohort was 10 (7.5–11).

Among a cohort of 434 patients, 37.7% (n = 164) and 6.6% (n = 29) individuals were fully and partially vaccinated, respectively, with 55.5% (n = 241) of the unvaccinated individuals.

The fully vaccinated individuals were older than the partially vaccinated and the unvaccinated individuals (66 vs 59 vs 55 years, p-value <0.001). A higher proportion of the patients with hypertension were fully vaccinated as compared to partially and unvaccinated individuals (58.5% vs 48.3% vs 43.2%, p-value 0.010). However, in patient who had chronic kidney disease, the proportion of patients with complete vaccination was lower as compared to partially and unvaccinated individuals (3.7% vs 13.8% vs 8.3%, p-value 0.045). There was no difference observed in the presenting symptoms and CALL score of the study individuals based on the vaccination status. see Table 1 for clinical characteristics of study population.

Majority (35.7%) of the patients had received Sinopharm vaccination as shown in Table 2 and the median time interval from the last dose of vaccination to symptom onset was 74 (42–114) days.

**Table 1. The clinical characteristics of the study population (N = 434).**

| | Total%(n = 434) | Fully Vaccinated%(n = 164) | Partially vaccinated%(n = 29) | Unvaccinated%(n = 241) | p-value** |
|---|---|---|---|---|---|
| **Age, years*** | 61(44–70) | 66(56–74) | 59(50–66) | 55(37.5–67) | 0.000 |
| **Gender** | | | | | |
| Male | 54.1(235) | 56.7(93) | 44.8(13) | 129(53.5) | 0.476 |
| Female | 45.9(199) | 43.3(71) | 55.2(16) | 112(46.5) | |
| **Hypertension** | | | | | |
| Yes | 49.3(214) | 58.5(96) | 48.3(14) | 43.2(104) | 0.010 |
| No | 50.7(220) | 41.5(68) | 51.7(15) | 56.8(137) | |
| **Ischemic heart disease** | | | | | |
| Yes | 15(65) | 16.5(27) | 20.7(6) | 13.3(32) | 0.398 |
| No | 85(309) | 83.5(137) | 79.3(23) | 86.7(209) | |
| **Diabetes mellitus** | | | | | |
| Yes | 43.8(190) | 46.3(76) | 51.7(15) | 41.6(99) | 0.387 |
| No | 56.2(244) | 53.7(88) | 48.3(14) | 58.9(142) | |
| **Chronic obstructive airway disease** | | | | | |
| Yes | 2.1(9) | 1.8(3) | (0) | 2.5(6) | 0.864 |
| No | 97.9(425) | 98.2(170) | 100(29) | 97.5(235) | |
| **Congestive heart failure** | | | | | |
| Yes | 1.8(8) | 2.4(4) | (0) | 1.7(4) | 0.840 |
| No | 98.2(426) | 97.6(160) | 100(29) | 98.3(237) | |
| **Asthma** | | | | | |
| Yes | 5.1(22) | 6.7(11) | 3.4(1) | 4.1(10) | 0.534 |
| No | 412(94.9) | 93.3(153) | 96.6(28) | 95.9(231) | |
| **Smoking** | | | | | |
| Yes | 1.6(7) | 2.4(4) | 3.4(1) | 0.8(2) | 0.172 |
| No | 98.4(427) | 97.6(160) | 96.6(28) | 99.2(239) | |
| **Cerebrovascular accident** | | | | | |
| Yes | 3.9(17) | 3 (5) | 10.3(3) | 3.7(9) | 0.171 |
| No | 96.1(417) | 97(159) | 89.7(26) | 96.3(232) | |
| **Chronic liver disease** | | | | | |
| Yes | 3(13) | 2.4(4) | (0) | 3.7(9) | 0.658 |
| No | 97(421) | 97.6(160) | 100(29) | 96.3(232) | |
| **Chronic kidney disease** | | | | | |
| Yes | 6.9(30) | 3.7(6) | 13.8(4) | 8.3(26) | 0.045 |
| No | 93.1(404) | 96.3(158) | 86.2(25) | 91.7(221) | |
| **≥2 Co-morbid conditions** | | | | | |
| Yes | 28.1(122) | 31.1(51) | 31(9) | 25.7(62) | 0.489 |
| No | 71.9(312) | 68.9(113) | 69(20) | 74.3(179) | |
| **Fever** | | | | | |
| Yes | 79.7 (346) | 79.3(130) | 89.7(26) | 78.8(190) | 0.385 |
| No | 20.3(88) | 20.7(34) | 10.3(3) | 21.2(51) | |
| **Cough** | | | | | |
| Yes | 60.8(264) | 63.4(104) | 69(20) | 58.1(140) | 0.363 |
| No | 39.2(170) | 36.6(60) | 31(9) | 41.9(101) | |
| **Dyspnea** | | | | | |
| Yes | 68.9(299) | 65.2(107) | 69(20) | 57.5(172) | 0.426 |
| No | 31.1(135) | 34.(57) | 31(9) | 51.5(69) | |
| **Sore throat** | | | | | |

*(Continued)*

**Table 1.** (Continued)

|  | Total%(n = 434) | Fully Vaccinated%(n = 164) | Partially vaccinated%(n = 29) | Unvaccinated%(n = 241) | p-value** |
|---|---|---|---|---|---|
| **Yes** | 4.4(19) | 6.1(10) | 3.4(1) | 3.3(8) | 0.394 |
| **No** | 95.6(415) | 93.6(154) | 96.6(28) | 96.7(233) |  |
| **GI symptoms** |  |  |  |  |  |
| **Yes** | 13.6(59) | 12.2(20) | 24.1(7) | 13.3(32) | 0.219 |
| **No** | 86.4(375) | 87.8(144) | 75.9(22) | 86.7(209) |  |
| **CALL score*** | 10(7.5–11) | 10(8–12) | 10(7–11) | 9(7–11) | 0.124 |

*Median (Interquartile ranges).

## Laboratory parameters of patients with COVID-19 infection in correlation to vaccination status

In comparison to the fully vaccinated group, the unvaccinated and partially vaccinated groups had considerably higher median serum procalcitonin, ferritin, and LDH levels (with statistically significant p-value). D-dimer levels were also significantly higher in the unvaccinated group when compared with partially and fully vaccinated groups. However, no differences were observed in serum levels of C-reactive protein (CRP) and troponin levels. See Table 3 for lab parameters.

Unvaccinated patients with severe/critical Covid-19 disease had significantly higher levels of median serum ferritin (p-value 0.001), LDH (p-value 0.013) and D-dimer levels (p-value 0.010) when compared with vaccinated patients with severe disease. However, no differences in median serum procalcitonin levels were observed among vaccinated and unvaccinated patients with severe/critical disease. Among patients with non-severe Covid-19, no significant differences in laboratory parameters were seen based on vaccination status. see Table 4.

## Computed tomography chest findings of patients with COVID-19 infection in relation to vaccination status

CT scan chest was done in 16.6% (n = 72) of the patients. Among these, 87.5%(n = 63) were CT pulmonary angiogram (CTPA). The median time interval between CT scan and admission

**Table 2. Type of Vaccination received by study participants.**

| Vaccine type | n (%) |
|---|---|
| **AstraZeneca AZD1222, ChAdOx1 nCoV-19** | 5 (1.2) |
| **Cansino AD5-nCOV** | 6(1.4) |
| **Moderna mRNA-1273** | 1(0.2) |
| **Pfizer BNT162b2** | 3(0.7) |
| **Sinopharm BBIBP-CorV** | 155(35.7) |
| **Sinovac PiCoVacc** | 22(5.1) |
| **Sputnik Gam-COVID-Vac** | 1(0.2) |
| **Total** | 193(44.5) |

**Table 3. Baseline laboratory parameters of patients with COVID-19 infection in correlation to vaccination status.**

| Variable | Total (n = 434) Median (IQR) | Fully vaccinated (n = 164) Median (IQR) | Partially vaccinated (n = 29) Median (IQR) | Unvaccinated (n = 241) Median (IQR) | p-value* |
|---|---|---|---|---|---|
| **Total leukocyte count**(x10^9/L) | 8(5.8–11.575) | 7.7(5.45–10.60) | 8.3(5.5–12) | 8.5(5.8–12.4) | 0.144 |
| **Creatinine** (mg/dl) | 1.1(0.8–1.475) | 1.1(0.825–1.4) | 1.05(0.725–1.5) | 1.1(0.8–1.5) | 0.570 |
| **C-Reactive Protein (mg/L)** | 80(31–141) | 82.85(33.25–145) | 112(32.4–172.65) | 71(29–135) | 0.261 |
| **Procalcitonin**(ng/ml) | 0.1795(0.081–0.47375) | 0.13(0.70–0.3215) | 0.2595(0.0915–0.6875) | 0.21(0.088–0.58) | 0.004 |
| **Ferritin**(ng/ml) | 467(208–1100) | 385(179.5–809) | 512(319–1033.5) | 612(216–1331.5) | 0.010 |
| **Lactate Dehydrogenase**(I.U/L) | 400.5(309.5–519.5) | 367(288–453) | 415(353.5–506) | 419(323–577.5) | 0.001 |
| **Trop I**(ng/L) | 0.0315(0.006–4) | 0.05(0.006–4) | 0.12(0.006–8) | 0.255(0.006–2) | 0.265 |
| **D-dimer**(mg/ml) | 1(0.5–2.4) | 0.80(0.50–1.5) | 0.8(0.55–3.5) | 1.2(0.6–3) | 0.005 |

*Independent sample Kruskal Wallis test.

date was 9 days (IQR 4–14). In 72 patients in which CT scan was done, around 69.4% (n = 50) of the patients had critical disease, 22.2% (n = 16) had severe disease and 8.3% (n = 6) patients had non-severe disease.

Most common finding was ground-glass opacities 91.7% (n = 66), followed by consolidation in 40.3% (n = 29). There were no statistically significant differences in CT severity score based on vaccination status. see Table 5.

## Comparison of clinical outcomes in Covid-19 infection in relation to the vaccination status

Among a cohort of 434 patients, the majority (58.5%) of the patients were admitted to HDU with 2.3% of the patient's requiring admission to the ICU. Around 3% and 42.9% of the patients required inotropic and NIV support respectively; however, there was no difference observed with regards to their vaccination status. In the case of unvaccinated patients, there were an increased number of HDU and ICU admissions with a statistically significant p-value. The severe (33.2% vs 34.5% vs 30.5%) and critical disease (44.8% vs 48.3% vs 34.8%) was significantly higher in the unvaccinated and partially vaccinated group as compared to the vaccinated group (p-value 0.04). Comparatively higher number of unvaccinated and partially vaccinated patients required invasive ventilation than the fully vaccinated cohort (7.9% vs 3.4% vs 1.8%, p-value 0.025). The majority (80.2%, n = 348) of the patients were discharged and 7.6% (n = 33) of the patient were discharged against medical advice. Overall, mortality rate was also significantly higher in unvaccinated patients(16.2%, n = 39) as compared to 6.1% (n = 10) in fully vaccinated patients (p-value 0.006). A similar trend was noticed for in-hospital mortality. There were no statistically significant differences in length of hospital stay, and in the median time interval from last vaccination dose among survivors and non-survivors (p-value 0.607). see Table 6.

Among complications, overall, acute kidney injury was the most common complication observed (24%, n = 104), followed by superimposed bacterial infection and sepsis in 15.4% (n = 67) and 13.4% (n = 58) respectively. Unvaccinated patients had significantly higher rate of sepsis (19.5% vs 6.7% p-value <0.001), septic shock (7.5% vs 0.6% p-value 0.002) and multi-organ dysfunction (9.1% vs 1.2% p-value 0.002) as compared to fully vaccinated patients. Pulmonary embolism was reported in 2.5% (n = 11) of the patients, however no statistically significant difference was observed based on vaccination status. Refer to Table 7 for clinical complications.

**Table 4. Baseline laboratory parameters of patients with COVID-19 infection in correlation to severity of disease and vaccination status.**

| Variable | Severe Disease/Critical disease | | | | | Non-Severe disease | | | | |
|---|---|---|---|---|---|---|---|---|---|---|
| | Total (n = 319) Median (IQR) | Fully vaccinated (n = 107) Median (IQR) | Partially vaccinated (n = 24) Median (IQR) | Unvaccinated (n = 188) Median (IQR) | p-value* | Total (n = 115) Median (IQR) | Fully vaccinated (n = 57) Median (IQR) | Partially vaccinated (n = 5) Median (IQR) | Unvaccinated (n = 53) Median (IQR) | p-value |
| **Total leukocyte count** (x10^9/L) | 9(6.1–12.5) | 9.0(6.3–11.5) | 8.75 (6.2–13.3) | 9.3(6–13.0) | 0.73 | 6.3(4.8–8.4) | 6.0 (4.8–8.1) | 5.7(4.9–10) | 6.7(4.8–9.1) | 0.57 |
| **C-Reactive Protein (mg/L)** | 106(45–151) | 110.0 (53–163) | 112.5 (38–172) | 101.(42–146) | 0.35 | 30(14–61) | 38.75(16.5–74.7) | 111.8(24.5–169) | 25.63(9.5–44.5) | 0.02 |
| **Procalcitonin** (ng/ml) | 0.21(0.92–0.55) | 0.17(0.09–0.36) | 0.35 (0.09–0.70) | 0.22(0.09–0.62) | 0.05 | 0.89(0.04–0.24) | 0.08(0.04–0.15) | 0.11(0.02–0.61) | 0.09(0.03–0.41) | 0.77 |
| **Ferritin** (ng/ml) | 567(282–1267) | 456.5 (227–1101) | 473.5 (316–1088) | 770.9(340–1382) | 0.01 | 252(94.7–534) | 283(118–470) | 689.7(369–975) | 208(52–612) | 0.14 |
| **Lactate Dehydrogenase** (I.U/L) | 424(353–559) | 388(325–492) | 439(355–525) | 457(368–603) | 0.00 | 293(232–377) | 306(247–379) | 336(298–441) | 271(225–377) | 0.27 |
| **D-Dimer**(mg/ml) | 1.2(0.6–2.7) | 1(0.6–1.5) | 1 (0.6–3.7) | 1(0.7–3.3) | 0.00 | 0.6(0.4–1.3) | 0.6(0.4–1.3) | 0.6(0.35–2.5) | 0.6(0.4–1.2) | 0.91 |
| **Trop I** (ng/L) | 0.047(0.006–6.0) | 0.052 (0.006–6.25) | 1.5 (0.01–8.50) | 0.040(0.006–4.00) | 0.30 | 0.006(0.0075–2.0) | 0.006(0.006–2.0) | 0.006(0.006–5.005) | 0.006(0.006–0.047) | 0.06 |
| **Creatinine** (mg/dl) | 1.2(0.9–1.5) | 1.1(0.9–1.5) | 1.1(0.7–1.6) | 1.2(0.9–1.5) | 0.47 | 0.9(0.8–1.3) | 1.0(0.8–1.3) | 0.9(0.7–1.2) | 0.9(0.7–1.1) | 0.35 |

**Table 5. Computed tomography chest findings of COVID-19 infection in vaccinated as compared to non-vaccinated patients.**

| | Total %(n = 72) | Fully Vaccinated %(n = 28) | Partially vaccinated %(n = 5) | Unvaccinated %(n = 39) | p-value |
|---|---|---|---|---|---|
| **CT severity score** | | | | | |
| **Mild** | 25(18) | 25(7) | 1.4(1) | 25.6 (10) | 0.993 |
| **Moderate** | 30.6(22) | 32.1 (9) | 40.0(2) | 28.2(11) | |
| **Severe** | 44.4(32) | 42.9(12) | 40.0(2) | 46.2(18) | |
| **Ground-glass haze/opacities** | | | | | |
| **Yes** | 91.7(66) | 92.9(26) | 100(5) | 89.7(35) | 0.808 |
| **No** | 8.3(6) | 7.1(2) | 100(29) | 10.3(4) | |
| **Pneumomediastinum** | | | | | |
| **Yes** | 5.6(4) | 3.6(1) | 0 | 7.7(3) | 0.728 |
| **No** | 94.4(68) | 96.4(27) | 100(5) | 92.3(36) | |
| **Pneumothorax** | | | | | |
| **Yes** | 4.2(3) | 3.6(1) | 0 | 5.1(2) | 0.847 |
| **No** | 95.8(69) | 96.4(27) | 100(5) | 94.9(37) | |
| **Mediastinal Lymphadenopathy** | | | | | |
| **Yes** | 12.5(9) | 3.6(1) | 0 | 20.5(8) | 0.093 |
| **No** | 87.5(63) | 96.4(27) | 100(5) | 79.5(31) | |
| **Effusion** | | | | | |
| **Yes** | 5.6(4) | 3.6(1) | 0 | 7.7(3) | 0.312 |
| **No** | 94.4(68) | 96.4(27) | 100(5) | 92.3(36) | |
| **Atelectasis** | | | | | |
| **Yes** | 6.9(5) | 7.1(2) | 0 | 7.7(3) | 1.00 |
| **No** | 93.1(67) | 92.9(26) | 100(5) | 92.3(36) | |
| **Consolidation** | | | | | |
| **Yes** | 40.3(29) | 35.7(10) | 40.0(2) | 43.6(17) | 0.863 |
| **No** | 59.7(43) | 64.3(18) | 60.0 (3) | 56.4(22) | |
| **Fibrosis** | | | | | |
| **Yes** | 34.7(25) | 39.3(811) | 40.0(2) | 30.8(12) | 0.683 |
| **No** | 65.3(47) | 60.7(17) | 60.0 (3) | 57.4(27) | |
| **Predominant pattern** | | | | | |
| **GGO** | 56.9(41) | 67.9(19) | 40(2) | 51.3(20) | |
| **Consolidation** | 5.6(4) | 10.7(3) | 0(0) | 2.6(1) | 0.088 |
| **Fibrosis** | 8.3(6) | 10.7(3) | 0(0) | 7.7(3) | |
| **Mixed** | 29.2(21) | 10.7(3) | 60(3) | 38.5(15) | |
| **Laterality** | | | | | |
| **Unilateral** | 4.2(3) | 0(0) | 00) | 7.7(3) | 0.405 |
| **Bilateral** | 95.8(69) | 100(28) | 100 (5) | 92.3(36) | |
| **Lobar involvement** | | | | | |
| **Unilobar** | 2.8(2) | 0(0) | 0(0) | 5.1(2) | 0.748 |
| **Bilobar** | 4.2(3) | 3.6(1) | 0(0) | 5.1(2) | |
| **Multilobar** | 93.1(67) | 96.4 (27) | 100 (5) | 89.7(35) | |

On multivariable regression, after adjusting for age, gender, and presence of ≥2 co-morbid conditions, vaccination status was an independent predictor of mortality and unvaccinated patients had statistically significant mortality risk with p-value <0.001 (OR 5.04, CI 2.04–10.55). Refer to Table 8 for predicators of mortality.

**Table 6. Clinical Outcomes of COVID-19 infection in vaccinated as compared to non-vaccinated patients.**

|  | Total %(n = 434) | Fully Vaccinated %(n = 164) | Partially vaccinated %(n = 29) | Unvaccinated %(n = 241) | p-value |
|---|---|---|---|---|---|
| **Level of care** |  |  |  |  |  |
| **Ward** | 39.2(170) | 46.3(76) | 41.4(12) | 34(82) | 0.043 |
| **HDU** | 58.5(254) | 53(87) | 58.6(17) | 62.2(150) |  |
| **ICU** | 2.3(10) | 0.6(1) | 0 | 3.7(9) |  |
| **Inotrope requirement** |  |  |  |  |  |
| **Yes** | 3(13) | 2.4(4) | 0 | 3.7(9) | 0.535 |
| **No** | 97(421) | 97.6(160) | 100(29) | 96.3(232) |  |
| **NIV requirement** |  |  |  |  |  |
| **Yes** | 42.9(186) | 37.8(62) | 44.8(13) | 46.1(111) | 0.251 |
| **No** | 57.1(248) | 62.2(102) | 55.2(16) | 53.9(130) |  |
| **COVID-19 Severity** |  |  |  |  |  |
| **Non-Severe** | 26.5(115) | 34.8(57) | 17.2(5) | 22(53) |  |
| **Severe** | 32.7(140) | 30.5(50) | 34.5(10) | 33.2(80) | 0.040 |
| **Critical** | 41.2(179) | 34.8(57) | 48.3(14) | 44.8(108) |  |
| **Intubation** |  |  |  |  |  |
| **Yes** | 5.3(23) | 1.8(3) | 3.4(1) | 7.9(19) | 0.025 |
| **No** | 94.7(411) | 98.2(161) | 96.6(28) | 92.1(222) |  |
| **Outcome** |  |  |  |  |  |
| **Survived & Discharged** | 80.2(348) | 88.4(145) | 86.2(25) | 73.9(178) | 0.004 |
| **Dead** | 12.2(53) | 6.4(10) | 13.8(4) | 16.2(39) |  |
| **LAMA** | 7.6(33) | 5.5(9) | 0 | 10(24) |  |
| **In-hospital Mortality** |  |  |  |  |  |
| **Yes** | 12.2(53) | 6.1(10) | 13.8(4) | 16.2(39) | 0.006 |
| **No** | 87.8(381) | 93.9(154) | 86.2(25) | 83.8(202) |  |
| **Length of stay** | 4(2.75–8) | 4(3–7) | 4(2.5–8.5) | 4(2–8) | 0.589 |

* HDU- High Dependency Unit; ICU- Intensive Care Unit; LAMA- Left against Medical Advice.

## Discussion

In Pakistan, the frequency and severity of breakthrough infection in vaccinated patients and especially the infection that leads to hospitalization and mortality needs to be evaluated. The findings of our study showed lower mortality and ICU admission rate in vaccinated patients.

Studies on vaccination effectiveness worldwide have been mostly based on real-world surveillance data. Reported data have been similar and showed reduced hospitalization and mortality risks. Haas EJ et al. [9] conducted a national surveillance study in Israel between Jan 24 to April 3, 2021, and showed that around 4481 COVID-19-related severe or critical hospitalizations occurred in people above 16 years of age. Among them 71.4% (3201) were unvaccinated and 8.1% (364) were fully vaccinated. The reported mortality was 64.2% and 12.4% in unvaccinated patients and vaccinated patients, respectively. Among the vaccinated patients, the severe/critical disease and mortality were higher in old, aged patients. Another surveillance study conducted in Australia demonstrated that around 61,800 patients suffered from COVID-19 infection from 16 June to 7 October 2021 and the vast majority (63.1%) of them were unvaccinated. Most of the patients who died were unvaccinated, and the mortality among fully vaccinated patients was 5.8%. Vaccination did decrease the risk of serious infection and death, and patients who died had an average age of 82 years with multimorbidity

**Table 7. Clinical complications of COVID-19 infection in vaccinated as compared to non-vaccinated patients.**

| | Total%(n = 434) | Fully Vaccinated%(n = 164) | Partially vaccinated%(n = 29) | Unvaccinated%(n = 241) | p-value |
|---|---|---|---|---|---|
| **Sepsis** | | | | | |
| Yes | 13.4(58) | 6.7(11) | 0 | 19.5(47) | 0.000 |
| No | 86.6(376) | 93.3(153) | 100(29) | 80.5(194) | |
| **Septic shock** | | | | | |
| Yes | 4.4(19) | 0.6(1) | 0 | 7.5(18) | 0.002 |
| No | 95.6(415) | 99.4(163) | 100(29) | 92.5(223) | |
| **Multi-organ dysfunction** | | | | | |
| Yes | 5.8(25) | 1.2(2) | 3.4(1) | 9.1(22) | 0.002 |
| No | 94.2(409) | 98.8(162) | 96.6 (28) | 90.9(219) | |
| **Pneumomediastinum** | | | | | |
| Yes | 2.1(9) | 1.8(3) | 0 | 2.5(6) | 0.864 |
| No | 97.9(425) | 98.2(161) | 100(29) | 97.5(235) | |
| **Pneumothorax** | | | | | |
| Yes | 12.2(8) | 2.4(4) | 0 | 1.7(4) | 0.840 |
| No | 87.8(426) | 97.6(160) | 100(29) | 98.3(237) | |
| **Arrhythmias** | | | | | |
| Yes | 7.4(32) | 9.1(15) | 10.3(3) | 5.8(14) | 0.312 |
| No | 92.6(402) | 90.9(149) | 89.7 (26) | 94.2(227) | |
| **Fungal infections** | | | | | |
| Yes | 9.4(41) | 7.3(12) | 6.9(2) | 11.2(21) | 0.418 |
| No | 90.6(393) | 92.7(152) | 93.1 (27) | 88.8(214) | |
| **Bacterial infections/HAP** | | | | | |
| Yes | 15.4(67) | 14(23) | 17.2(5) | 16.2(39) | 0.774 |
| No | 84.6(434) | 86(141) | 82.8 (24) | 83.8(202) | |
| **Acute kidney injury** | | | | | |
| Yes | 24(104) | 19.5(32) | 17.2(5) | 27.8(67) | 0.114 |
| No | 76(330) | 80.5(132) | 82.8 (24) | 72.2(174) | |
| **Deranged liver function** | | | | | |
| Yes | 6.0(26) | 4.3(7) | 10.3(3) | 6.6(16) | 0.294 |
| No | 94.0(408) | 95.7(157) | 89.7 (26) | 93.4(225) | |
| **Diabetic ketoacidosis** | | | | | |
| Yes | 2.5(11) | 3(5) | 0 | 2.5(6) | 0.89 |
| No | 97.5(423) | 97(159) | 100(29) | 97.5(235) | |
| **Myocardial injury** | | | | | |
| Yes | 6.9(30) | 4.9(8) | 10.3(3) | 7.9(19) | 0.350 |
| No | 93.1(404) | 95.2(156) | 89.7 (26) | 92.1(222) | |
| **Pulmonary embolism** | | | | | |
| Yes | 2.5(11) | 1.8(3) | 6.9(2) | 2.5(6) | 0.192 |
| No | 97.5(423) | 98.2(161) | 93.1 (27) | 97.5(235) | |

[10]. However, these studies have not evaluated the in-hospital differences between the clinical and laboratory parameters among vaccinated and unvaccinated patients. Moreover, surveillance studies in a low-resource country such as Pakistan are confounded due to the lack of electronic health care systems and non-uniform health care setups. Results have been variable among hospitalized cohorts due to differences in vaccination types, no standardized definition of fully and partially vaccinated individuals, inadequate healthcare resources, heterogeneous populations, and differences in the time interval between vaccination and onset of

**Table 8. Multi-variable regression analysis of predictors of mortality.**

| Variable | p-value | Odds ratio(95% Confidence interval) |
|---|---|---|
| **Age>65 years** | 0.000 | 7.46(3.67–15.16) |
| **Male gender** | 0.649 | 1.15(0.61–2.17) |
| **More than 2 co-morbid conditions** | 0.024 | 2.08(1.1–3.96) |
| **Unvaccinated** | 0.000 | 5.04(2.04–10.55) |

also, Lower Odds for Fully vaccinated 0.19(0.095–0.41).

breakthrough infections. However, in comparison to these studies, our study has evaluated the in-hospital differences between the clinical and laboratory parameters among vaccinated and unvaccinated patients and has also used standardized definitions of fully and partially vaccinated individuals.

Hu et al. [11] from China reported a lower risk of progression to severe disease in a study on patients who received inactivated COVID-19 vaccine. Majority (73.3%) had received CoronaVac (Sinovac Biotech, Beijing, China), and 26.5% had received BBIBP-CorV (Sinopharm). They also reported lower lactate dehydrogenase (LDH) levels among vaccinated and no differences in CRP levels which is consistent with our study findings. It is possible that we didn't find any difference in the CRP level when comparing the vaccinated and unvaccinated patients as we only compared the admission CRP level. However, there is a possibility that the trend of CRP may be of more value in predicating the severity of disease among vaccinated and unvaccinated individuals.

Sagiraju HKR et al. [12] from India conducted a similar study and reported a lower mortality rate in vaccinated patients; however, only 3% of patients in their study cohort were completely vaccinated as compared to 37.7% patients in our study. Almost 15% of the patients with breakthrough infections were asymptomatic, dyspnea was reported in 22%, and a higher proportion of unvaccinated patients were symptomatic in their study. However, no differences in the severity of symptoms between vaccinated and unvaccinated individuals were observed in our study. Compared to vaccinated individuals, unvaccinated individuals had elevated levels of D-dimer, interleukin 6 (IL-6), ferritin, LDH, and CRP. These findings were similar to our study with the exception that we found no variation in CRP levels when vaccination status was taken into consideration. In both the studies, the vaccinated group as compared to the unvaccinated group had lesser odds of requiring oxygen/ventilatory support with progression to critical illness and death. Papagoras et al. from Greece [13] also reported better outcomes in vaccinated patients. As per Papagoras et al. supplemental oxygen requirement, NIV use, and mortality was higher in unvaccinated patients. But the population included in this study comprised of patients with systemic rheumatic disease.

Balachandran et al. [14] compared outcomes among vaccinated and unvaccinated in a retrospective study in South Kerala, India, and reported 4.21 timer higher odds of mortality among unvaccinated patients; similar to results reported in our study. They also reported a higher prevalence of respiratory and neurological symptoms in vaccinated patients; however, no such findings were noted in our study population.

Muthukrishnan et al. [15] from India also conducted a hospital-based cross-sectional study and reported a higher mortality rate of 31.45% vs 12.5% among unvaccinated as compared to those fully vaccinated. Moreover 70% lower risks of mortality were reported in the fully vaccinated cohort.

Butt et al. [16] from Qatar reported a significantly higher proportion of developing the severe disease among unvaccinated patients. Similar findings were observed in our study.

Lakhia et al. [17] compared the CT severity score of the completely vaccinated patients with unvaccinated individuals. On the multivariate linear regression model, the CT severity score was significantly higher in unvaccinated patients as compared to fully vaccinated patients. However, this is contradictory to our study as we found no statistically significant differences in CT severity score based on vaccination status. Lee et al [18] reported a higher proportion of CT scans without pneumonia in vaccinated patients as compared to unvaccinated patients. In agreement with our study, they also reported vaccinated patients having lower rates of ICU admission. Furthermore, no significant differences in LOS, CRP, and LDH levels were noted between the three patient categories in their study. In our study, the predominant radiological pattern in the vaccinated patients was presence of ground glass opacities, whereas mixed pattern was predominant pattern in unvaccinated patients. This was also observed in the study done by Verma et al. [19] and he also reported similar rates of effusion, fibrosis, atelectasis, and mediastinal lymphadenopathy based on vaccination status.

In a register-based cohort study of 3,203 patients in Norway by Whittaker et al. [20], fully vaccinated patients had a lower risk of ICU admission which is similar to our study; however, they observed no differences in mortality among vaccinated and unvaccinated patients. Although there were no significant differences in LOS among vaccinated and unvaccinated in our study, shorter LOS among vaccinated patients was reported in their study.

Thompson et al. [21] in their study showed that the effectiveness of vaccination against COVID-19 infection leading to an ICU and emergency department admission or urgent care clinic visit was 90% (95% CI, 86 to 93) and 91% (95% CI, 89 to 93), respectively, with the effectiveness of BNT162b2 and mRNA-1273 vaccines from 81% to 95%. However, Ad26.COV2.S vaccine had an effectiveness of 68% to 73%. In Pakistan, many of the other vaccines such as BNT162b2 and mRNA-1273 vaccines were not available and only Sinopharm (BBIBP-CorV) was available. Due to this reason, we were not able to explore the effectiveness of several types of COVID-19 vaccines in our population.

In our study, pulmonary embolism was reported in 11/63 patients(17.4%), however no statistically significant difference was observed based on vaccination status. In a single center study, on the prevalence of pulmonary embolism in COVID-19 patients by Law et al. [22], unvaccinated patients had a 2.75-fold higher risk of pulmonary embolism as compared to vaccinated patients (p-value 0.02). CTPA were done in 18.3% patients as compared to 14.3% in our study. Hence, there is a probability that the percentage of pulmonary embolism could have been higher if CT scan were done in more patients and then there could have been a statistically significant difference between the vaccinated and unvaccinated population.

There are certain limitations of our study. This was a single-center observational study conducted in a large private tertiary care hospital; therefore, the results cannot be generalized to the entire population. Most of our patients had received Sinopharm (BBIBP-CorV) vaccine, hence a comparison between characteristics, clinical profiles, and outcomes of COVID-19 infection among different vaccines was not possible due to the smaller cohort of patients receiving other vaccines. To see the effectiveness of the vaccine, COVID-19 antibodies tests were not done. There were errors in documentation; however, all such gaps were covered by reviewing the patient's medical records thoroughly. Although baselines clinical characteristics and co-morbid conditions were similar across both groups, whether these conditions were controlled was not evaluated. A validated scoring system for illness severity was not used; however, a CALL score was used. The mortality in the unvaccinated patient was 16.2%; however, as we were not able to determine the outcome of patients who left against medical advice so this percentage can be higher. Despite these limitations, to our knowledge, this is the first reported data on outcomes based on vaccination status from Pakistan. The study has also

evaluated the differences in radiological features and complications among vaccinated and unvaccinated cohorts.

## Conclusion

Almost two years have passed and yet there is no effective therapy (with minimal side effects) to treat COVID-19 infection. To prevent COVID-19 infection from causing severe and critical illness, prevention through vaccination is our only hope, and to understand the prevalence and mechanism of breakthrough infection in vaccinated individuals, more research needs to be done. However, through this preliminary data, we can conclude that patients do develop severe and critical COVID-19 infection despite being vaccinated. However, the proportion of severe COVID-19 infection, the requirement of ventilatory support, and overall mortality is lower as compared to the unvaccinated population. So, uninterrupted endeavors need to be done to vaccinate as many individuals as possible and necessity of booster doses also needs to be assessed. Furthermore, a more effective vaccination must be produced to lessen the high death toll of COVID-19 infection.

## Supporting information

**S1 File.**
(SAV)

## Author Contributions

**Conceptualization:** Samar Fatima, Amara Zafar, Haris Afzal, Taymmia Ejaz, Sara Shamim.

**Data curation:** Samar Fatima, Amara Zafar, Haris Afzal, Taymmia Ejaz, Sara Shamim, Shayan Saleemi, Amna Subhan Butt.

**Formal analysis:** Samar Fatima, Haris Afzal, Taymmia Ejaz, Amna Subhan Butt.

**Methodology:** Samar Fatima, Amara Zafar, Haris Afzal, Taymmia Ejaz, Shayan Saleemi.

**Project administration:** Samar Fatima, Amara Zafar.

**Software:** Taymmia Ejaz, Sara Shamim.

**Supervision:** Samar Fatima, Amara Zafar, Amna Subhan Butt.

**Validation:** Samar Fatima.

**Visualization:** Amna Subhan Butt.

**Writing – original draft:** Samar Fatima, Amara Zafar, Shayan Saleemi.

**Writing – review & editing:** Amara Zafar, Haris Afzal, Taymmia Ejaz, Sara Shamim, Amna Subhan Butt.

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
