## [Decision Letter · Decision Letter 0]

5 May 2022

PONE-D-22-06757COVID-19 infection among vaccinated and unvaccinated: Does it make any difference?PLOS ONE

Dear Dr. Samar Fatima,

Thank you for submitting your manuscript to PLOS ONE. After careful consideration, we feel that it has merit but does not fully meet PLOS ONE’s publication criteria as it currently stands. Therefore, we invite you to submit a revised version of the manuscript that addresses the points raised during the review process.

Please refer to the reviewer's comments and address the same.

We look forward to receiving your revised manuscript.

Kind regards,

Yatin N. Dholakia, MD

Academic Editor

PLOS ONE

Journal Requirements:

3. In your statement, please include the full name of the IRB or ethics committee who approved or waived your study, as well as whether or not you obtained informed written or verbal consent. If consent was waived for your study, please include this information in your statement as well.

Additional Editor Comments (if provided):

A good concept analyzing the effect of vaccination on Covid19.

There are minor issues that need to be addressed as per the reiviewer's comments.

Reviewers' comments:

Reviewer's Responses to Questions

**Comments to the Author**

1. Is the manuscript technically sound, and do the data support the conclusions?

Reviewer #1: Yes

2. Has the statistical analysis been performed appropriately and rigorously? 

Reviewer #1: Yes

3. Have the authors made all data underlying the findings in their manuscript fully available?

Reviewer #1: Yes

4. Is the manuscript presented in an intelligible fashion and written in standard English?

Reviewer #1: Yes

5. Review Comments to the Author

Reviewer #1: 1. There are many grammatical errors, which need to be rectified eg.

a. “Partially vaccinated” was defined as when individuals became symptomatic two or more weeks after the first dose and not received the second dose or received the second dose less than two weeks before getting symptomatic

b. A higher proportion of the patients with hypertension were fully vaccinated than partially vaccinated and the unvaccinated (58.5% vs 48.3% vs 43.2%, p-value 0.010)

c. However, the proportion of patients with fully vaccinated was lower than partially vaccinated and unvaccinated in patients having chronic kidney disease (3.7% vs 13.8% vs 8.3%, p-value 0.045)

d. The proportions of those with high D-Dimer, interleukin 6 (IL-6), ferritin, lactate dehydrogenase (LDH), and C-reactive protein (CRP) were significantly lower in the vaccinated patients as compared to the unvaccinated population and these findings are equivalent to our study except that we did not find any difference in CRP level based on vaccination status.

e. Almost 2 years have passed and yet the therapy effective by enlarge with minimum adverse effects is awaited to treat COVID-19 infection

2. “Unvaccinated patients had significantly higher median serum procalcitonin, ferritin, LDH and D-Dimer levels”…..Was it because they had more severe disease ? Please comment and clarify in the text.

D-Dimer level were significantly higher in the unvaccinated group when compared with partially vaccinated and fully vaccinated. However, no differences were observed in serum levels of C-Reactive protein and troponin levels.

This is odd as we have been observing high CRP and monitoring the same, and not prolactin. Any references for the same ?

3. Out of the 72 CT scans performed, how many were critical patients ? Were CT scans performed in asymptomatic patients as well or were all 72 patients critical ?

4. Was CTPA done in any of the patients ? Was PTE observed in any patient ? We know that COVID 19 is a prothrombotic state and PTE has been reported commonly. It would be interesting if this was also observed in unvaccinated individuals.

5. CT severity score definitions of mild, moderate, severe to be mentioned.

6. The clinical and laboratory data have been correlated with vaccinated vs unvaccinated individuals BUT how many critical patients had deranged laboratory parameters ? How many required ICU stay ? Essentially, deranged laboratory parameters are possibly due to more severe disease in the unvaccinated. Please mention if so.

7. Lymphopenia is a common finding in COVID 19 ? Was the occurrence of this noted ? If yes, did it have any bearing on worsening inspite of vaccination ? Unvaccinated patients had significantly higher rate of sepsis (19.5% vs 6.7% p-value… was this due to lymphopenia ?

8. Were the spike protein antibody levels done ? If yes, was there any correlation between time since vaccination and the levels and antibody levels and the severity of infection?

9. Which wave was this…first, second ? Driven by which variant ?

10. Correlation between type of vaccine and occurrence of breakthrough, severity of infection and mortality ?

11. They also reported lower LDH levels among vaccinated and no differences in CRP levels which is consistent with our study’s findings. What is the explanation for this ? Is it related to severity of infection ?

12. However, the proportion of patients with fully vaccinated was lower than partially vaccinated and unvaccinated in patients having chronic kidney disease (3.7% vs 13.8% vs 8.3%, p-value 0.045)……Any particular reason for this OR was this just an observation ?

6. PLOS authors have the option to publish the peer review history of their article (what does this mean?). If published, this will include your full peer review and any attached files.

Reviewer #1: No

---

## [Author Response · Author response to Decision Letter 0]

20 May 2022

Thank you for your reply regarding our manuscript # PONE-D-22-06757; entitled “COVID-19 infection among vaccinated and unvaccinated: Does it make any difference?”.

We are grateful for your and the reviewer’s comments. The manuscript is reviewed and modified according to the reviewer’s comments and all suggestions from reviewers are incorporated. We hope that our modifications render our manuscript in its current form suitable for publication in PLOS ONE. We look forward to hearing from you regarding our submission. We would be glad to respond to any further questions and comments that you may have.

Yours sincerely,

On behalf of all the authors,

Samar Fatima

Following are the responses to the comments of the academic editor and reviewers.

Academic editor:

Comment 1:

Reply: 

Thanks for highlighting this aspect. The current revised version is according to the PLOS ONE’S style requirement.

Comment 2:

Please provide additional details regarding participant consent. In the ethics statement in the Methods and online submission information, please ensure that you have specified what type you obtained (for instance, written or verbal, and if verbal, how it was documented and witnessed). If your study included minors, state whether you obtained consent from parents or guardians. If the need for consent was waived by the ethics committee, please include this information.

Reply: 

This was a prospective, observational study conducted by reviewing medical charts and electronic data. It did not involve any live interview or any direct interaction with the patients. Patients' confidentiality and anonymity were maintained, no identifiers that can be used to track participants were utilized, and the research questionnaire was identified by a serial number. The study was approved as an exemption by the ethical review committee (ERC) of Aga Khan University Hospital (AKUH), Karachi, Pakistan (IRB reference number: 2021-6478-18343). 

This statement we have modified and included in the manuscript with track changes.

Comment 3:

In your statement, please include the full name of the IRB or ethics committee who approved or waived your study, as well as whether or not you obtained informed written or verbal consent. If consent was waived for your study, please include this information in your statement as well.

Reply: 

Please review the reply for comment number 2. As per your suggestion we have modified our manuscript and included the IRB committee’s name and IRB number.

Comment 4:

Please review your reference list to ensure that it is complete and correct. If you have cited papers that have been retracted, please include the rationale for doing so in the manuscript text or remove these references and replace them with relevant current references. Any changes to the reference list should be mentioned in the rebuttal letter that accompanies your revised manuscript. If you need to cite a retracted article, indicate the article’s retracted status in the References list and include a citation and full reference for the retraction notice.

Reply: 

We have rechecked the references. There are no articles in the references that have been retracted. 

Reviewer # 1:

Comment 1: 

There are many grammatical errors, which need to be rectified eg.

a. “Partially vaccinated” was defined as when individuals became symptomatic two or more weeks after the first dose and not received the second dose or received the second dose less than two weeks before getting symptomatic

b. A higher proportion of the patients with hypertension were fully vaccinated than partially vaccinated and the unvaccinated (58.5% vs 48.3% vs 43.2%, p-value 0.010)

c. However, the proportion of patients with fully vaccinated was lower than partially vaccinated and unvaccinated in patients having chronic kidney disease (3.7% vs 13.8% vs 8.3%, p-value 0.045)

d. The proportions of those with high D-Dimer, interleukin 6 (IL-6), ferritin, lactate dehydrogenase (LDH), and C-reactive protein (CRP) were significantly lower in the vaccinated patients as compared to the unvaccinated population and these findings are equivalent to our study except that we did not find any difference in CRP level based on vaccination status.

e. Almost 2 years have passed and yet the therapy effective by enlarge with minimum adverse effects is awaited to treat COVID-19 infection

Reply: 

We are thankful to you for highlighting grammatical errors in our manuscript. The manuscript is revised accordingly. 

Comment 2: 

“Unvaccinated patients had significantly higher median serum procalcitonin, ferritin, LDH and D-Dimer levels” ….Was it because they had more severe disease ? Please comment and clarify in the text.

D-Dimer level were significantly higher in the unvaccinated group when compared with partially vaccinated and fully vaccinated. However, no differences were observed in serum levels of C-Reactive protein and troponin levels.

This is odd as we have been observing high CRP and monitoring the same, and not prolactin. Any references for the same?

Reply: 

Unvaccinated patients with severe/critical Covid-19 disease had significantly higher levels of median serum ferritin (0.001), LDH (0.013) and D-Dimer levels ( 0.010) when compared with vaccinated patients with severe disease. However, no differences in median serum procalcitonin levels were observed among vaccinated and unvaccinated patients with severe/critical disease.

Among patients with non-severe Covid-19, no significant differences in laboratory parameters were seen based on vaccination status. 

Overall, irrespective of vaccination status, patients with severe/critical disease had deranged laboratory parameters. We have attached an extensive table for your review, comparing these parameters based on the disease severity and the vaccination status. It is possible that severity of disease in unvaccinated patients was higher and resulted in these findings. See table for reference. This table has been added to the manuscript 

In reference to the CRP level, we may have not found any difference in the CRP level when comparing the vaccinated and unvaccinated patients as we only took the admission CRP level. However, there is a possibility that the trend of CRP may be of more value in predicating severity of disease among vaccinated and unvaccinated individuals. Similar results were reported by Hu et al. (reference: Hu Z, Tao B, Li Z, Song Y, Yi C, Li J, et al. Effectiveness of inactivated COVID-19 vaccines against severe illness in B.1.617.2 (Delta) variant–infected patients in Jiangsu, China. Int J Infect Dis [Internet]. 2022;116:204–9. Available from: https://doi.org/10.1016/j.ijid.2022.01.030). This statement has also been included in the discussion.

Comment 3: 

Out of the 72 CT scans performed, how many were critical patients? Were CT scans performed in asymptomatic patients as well or were all 72 patients critical?

Reply:

In 72 patients in which CT scan was done, around 69.4% (n=50) of the patients had critical disease, 22.2% (n=16) had severe disease and 8.3% (n=6) patients had non-severe disease. We have added this in the manuscript (with track changes).

Comment 4: 

Was CTPA done in any of the patients? Was PTE observed in any patient? We know that COVID 19 is a prothrombotic state and PTE has been reported commonly. It would be interesting if this was also observed in unvaccinated individuals.

Reply: 

CT scan chest was done in 72 patients, and among them CT pulmonary angiogram CTPA) was done in 87.5%(n=63) of the patients. Pulmonary embolism was reported in 11/63(17.4%) patients who underwent CTPA( overall 2.5%,n=11/434), however no statistically significant difference was observed based on vaccination status which could be due to smaller number of patients undergoing CT scan. Hence, there is a probability that the percentage of pulmonary embolism could have been higher if CT scan was done in more patients and then there could have been a statistically significant difference between the vaccinated and unvaccinated population. We have included this in our discussion and have also mentioned the relevant study in the discussion.

Comment 5: 

CT severity score definitions of mild, moderate, severe to be mentioned.

Reply: 

Based on the percentage involvement, individual lobar scoring was done, score ranged 1-5 respectively for involvement 5% or less, 5-25%,26%-49%,50-75% and >75% involvement. CT severity scoring was based on sum of all the individual lobar scores and categorized as Mild , score ≤7; 8-17 Moderate and ≥18 as Severe. We have included this in the manuscript as well.

Comment 6: 

The clinical and laboratory data have been correlated with vaccinated vs unvaccinated individuals BUT how many critical patients had deranged laboratory parameters? How many required ICU stay? Essentially, deranged laboratory parameters are possibly due to more severe disease in the unvaccinated. Please mention if so.

Reply: 

Please refer to comment number 2 for the answer for critical patients had deranged laboratory parameters. 

About the HDU/ICU stay around 58.5%(n=254) were admitted in HDU and around 2.3%(n=10) patients required ICU admission. This is mentioned in table 6 “Clinical outcomes of COVID-19 infection”. Among all, most of the patient who required HDU/ICU were unvaccinated with significant p-value. 

Comment 7: 

Lymphopenia is a common finding in COVID 19? Was the occurrence of this noted? If yes, did it have any bearing on worsening inspite of vaccination? Unvaccinated patients had significantly higher rate of sepsis (19.5% vs 6.7% p-value… was this due to lymphopenia?

Reply: 

We did not include lymphopenia in our study questionnaire because steroids increase lymphopenia and most patients had received steroids in the emergency department or in the outside clinics/hospital settings.

Comment 8: 

Were the spike protein antibody levels done ? If yes, was there any correlation between time since vaccination and the levels and antibody levels and the severity of infection?

Reply: 

To see the effectiveness of the vaccine spike protein antibody were not tested. This we have mentioned in the limitation of our study.

Comment 9: 

Which wave was this…first, second? Driven by which variant?

Reply: 

The study subjects were admitted between 1st June and 30th September. The 3rd wave in Pakistan began in mid-March and the 4th wave started in July. So, this data was collected by mid or end of 3rd wave and included patients from 4th wave as well. The 3rd and 4th waves were driven by 20I/501Y.V1, VOC 202012/01, or B.1.1.7 variant and B.1.617.2 or Delta variant respectively. 

Comment 10: 

Correlation between type of vaccine and occurrence of breakthrough, severity of infection and mortality?

Reply: 

Most of our patients had received Sinopharm (BBIBP-CorV) vaccine and the comparison between characteristics, clinical profiles, and outcomes of COVID-19 infection among different vaccines was not possible due to the smaller sample size of patients receiving other vaccines. Analysis of statistical significance to evaluate and compare these were not possible, given the very low percentage of patient receiving other vaccines. This has been mentioned in the limitations of our study.

Comment 11:

They also reported lower LDH levels among vaccinated and no differences in CRP levels which is consistent with our study’s findings. What is the explanation for this? Is it related to severity of infection?

Reply: 

See comment 2 

Comment 12:

However, the proportion of patients with fully vaccinated was lower than partially vaccinated and unvaccinated in patients having chronic kidney disease (3.7% vs 13.8% vs 8.3%, p-value 0.045) ……Any particular reason for this OR was this just an observation?

Reply: 

This was just an observation.

---

## [Decision Letter · Decision Letter 1]

12 Jun 2022

COVID-19 infection among vaccinated and unvaccinated: Does it make any difference?

PONE-D-22-06757R1

Dear Dr. Samar Fatima,

We’re pleased to inform you that your manuscript has been judged scientifically suitable for publication and will be formally accepted for publication once it meets all outstanding technical requirements.

Kind regards,

Yatin N. Dholakia, MD

Academic Editor

PLOS ONE

Additional Editor Comments (optional):

The reviewer has acknowledged that all comments have been satisfactorily addressed. There are few language edits that need to be made.

Reviewers' comments:

Reviewer's Responses to Questions

**Comments to the Author**

1. If the authors have adequately addressed your comments raised in a previous round of review and you feel that this manuscript is now acceptable for publication, you may indicate that here to bypass the “Comments to the Author” section, enter your conflict of interest statement in the “Confidential to Editor” section, and submit your "Accept" recommendation.

Reviewer #1: All comments have been addressed

2. Is the manuscript technically sound, and do the data support the conclusions?

Reviewer #1: Yes

3. Has the statistical analysis been performed appropriately and rigorously? 

Reviewer #1: Yes

4. Have the authors made all data underlying the findings in their manuscript fully available?

Reviewer #1: Yes

5. Is the manuscript presented in an intelligible fashion and written in standard English?

Reviewer #1: Yes

6. Review Comments to the Author

Reviewer #1: All the queries have been addressed satisfactorily. A few grammatical errors need to be addressed.

1. Line 315.....severe (instead of the severe)

2. Line 348...CT scans (instead of CT scan)

3. Line 357....baseline (instead of baselines)

4. Line 360....unvaccinated patients (instead of patient)

5. Line 366...yet there is no effective therapy.....this sentenced can be rephrased.

6. Line 375...a more effective vaccine (instead of vaccination) must be produced.

7. PLOS authors have the option to publish the peer review history of their article (what does this mean?). If published, this will include your full peer review and any attached files.

Reviewer #1: No

---

## [Editor Report · Acceptance letter]

8 Jul 2022

PONE-D-22-06757R1 

COVID-19 infection among vaccinated and unvaccinated: Does it make any difference? 

Dear Dr. Fatima:

I'm pleased to inform you that your manuscript has been deemed suitable for publication in PLOS ONE. Congratulations! Your manuscript is now with our production department. 

Kind regards, 

on behalf of

Dr. Yatin N. Dholakia 

Academic Editor

PLOS ONE